# Analysis of Liquid Quantity Measurement in Loading/Unloading Processes in Cylindrical Tanks

**Asta Meškuotienė, Paulius Kaškonas, Benas Gabrielis Urbonavičius, Gintautas Balčiūnas and Justina Dobilienė ***

Faculty of Electrical and Electronics Engineering, Institute of Metrology, Kaunas University of Technology, Studentų St. 50, LT-51368 Kaunas, Lithuania; asta.meskuotiene@ktu.lt (A.M.); paulius.kaskonas@ktu.lt (P.K.); benas.urbonavicius@ktu.lt (B.G.U.); gintautas.balciunas@ktu.lt (G.B.)
* Correspondence: justina.dobiliene@ktu.lt; Tel.: +370-37-351252

**Abstract:** Tanks, as instruments in oil and its product's amount measurement system chain, must be regularly maintained and metrologically inspected, as they significantly contribute to measurement uncertainty. However, when measuring a change in the amount of stored material (i.e., transfer), the measurement uncertainty becomes highly dependent not only on the mass of the transaction but also on the initial liquid level in the tank. This paper provides modeling of the uncertainties of the measuring system, which involves tanks, oil, and its product loading/unloading processes. It is shown that the accuracy of volume/mass measurement depends not only on the tank calibration table but also on the accuracy of other measuring instruments used and on the level of the liquid at the moment of measurement. The relative uncertainty of the measurement of the change in product mass depends linearly on the tank fill level present at the time of the transaction but nonlinearly on the transaction mass quantity.

**Keywords:** vertical cylindrical tank; volume/mass measurement; amount transfer; loading/unloading process; uncertainty





## 1. Introduction

Fixed storage tanks at atmospheric pressure or under pressure (hereinafter called "tanks") are built for bulk liquid storage and may be used for the measurement of quantities (volume or mass) of the liquid contained [1]. When operating fuel tanks, they are generally assigned to legal metrology and must meet some technical–metrological requirements [2,3], and they are used to measure products for custody transfer, leak detection, and inventory control [4,5]. Fuel quantity measurements are assigned to the most sensitive areas for public measurement [5,6]. Particular attention is being paid to this area [7,8], contributing to the development of a more efficient and society-friendly metrology system [9]. Thus, the fuel storage tank level measurements are constantly upgraded, including software [10,11]. Quality inventory control programs such as JIT (Just-in-time) [12], Lean Manufacturing [13], etc., are adopted for companies. Attention is paid to self-diagnostic and preventing overfill [3]. The new standards still allow the old practice to be maintained but offer better methods for more accurate measurement of oil volume and a much safer oil transfer process [3,14]. Therefore, tanks are subject to regular technical and metrological maintenance [15]; that is, they are calibrated at regular intervals from 5 to 15 years [3].

Storage tanks can contain large volumes of liquid products with significant monetary value [16,17], making the metrological control of fuel tanks very important. Any inaccuracy can cause problems in the supply chain. The use of known methods [18,19] and controlled measuring instrument characteristics allows for a valid fuel quantity assessment process [20]. Different parties in the performed commercial transactions agree on the different measuring instruments and methods used in the workflow and the interpretation of possible measurement errors [15]. Automated tank gauging and inventory control systems

are commonly used to manage tank farm operations [15,21,22]. Operationally reliable methods for measuring product quantity have been proposed [16,18,23]. The tools used allowed for the identification of the sources of errors in the measurement processes [24,25]. These factors minimize the impact of the mentioned sources and reduce losses in commercial transactions when they are used for measuring the absolute volume or mass at different levels [15,17,26].

A typical refinery fuel tank terminal can store up to 100 million liters of fuel. The 3% storage loss for such a system equates to 3 million liters of fuel [27]. The uncertainty of custody transfer has economic consequences, depending on the quantity of product transferred in a particular transaction [28]. Inefficient inventory control can be costly because it causes problems such as unplanned stock-outs, wasted resources, and inventory shrinkage. Inventory management directly or indirectly impacts inventory accuracy [29]. Most economic losses occur when tanks are regularly filled or emptied with large amounts of products. Thus, when operating large volume (100–60,000) $m^3$ tanks, the measurement of the absolute filling of the tank is not the sole solution for complete inventory control [17,30]. Considerable attention has been paid to the adjustment of the absolute measured value in the American Petroleum Institute (API), International Organization for Standardization (ISO) standards [2,3,14,31], and References [25,32]. The temperature or density change compensation required to accommodate changes in the level of the measured material due to thermal expansion does not eliminate this problem. The amount of product transferred at different levels is also important for tank operators. It is essential to understand the level rate at which a tank is filled or discharged. Are the uncertainties, measuring influencing factors, and measuring conditions affected by the tank fill level variation? Do these variations affect the monetary losses?

This study aimed to contribute to effective and accurate inventory solutions to maintain the balance between customer satisfaction and company revenue. It is important to reveal the criteria for reducing losses due to errors in volume/mass measurement during loading/unloading operations of a certain amount of product in the tank.

To achieve the aim, the following aspects are covered:

- Determination of the main factors of absolute quantity measurement uncertainty, using static volumetric assessments of the tank;
- Determination of the main factors of measurement uncertainty for the quantity change resulting from the loading/unloading processes;
- Modeling and comparison of absolute mass and mass difference measurement processes at the tank;
- Formulation of recommendations for improving the efficiency of fuel control and sales processes in a company related to quantity measurements in the tank.

It should be noted that the focus is not on the sources of systematic errors and their reduction but on the components of combined uncertainty, which are the qualitative characteristics of the measurement result.

## 2. Modeling of Material Mass Estimation in Vertical Tanks

Mathematical modeling was performed to evaluate the influence of tank calibration and factors on absolute mass and mass difference measurements. Modeling is required to reveal the contributions of the main uncertainty components to measurement accuracy in different measurement scenarios. This also allowed a comparison of the measurement results of the absolute mass and the difference in mass. Various tank-gauging techniques are used for refinery inventory control, stock accounting, and custody transfer applications [25,33,34]. In implementing these methods, most inventory control systems estimate inventory in real time, considering a variety of sources of errors. The oil industry often uses static volumetric assessments of tank contents [21,22]. The mass of the contained product was evaluated indirectly by measuring the fill level, tank geometric parameters, liquid temperature, and density.

Mass is calculated based on the following:

1. The tank calibration table that shows the dependency of the filling level and the volume of the contained liquid. If the tank is not thermally insulated, it is necessary to assess the change in the tank's volume due to thermal expansion [1,17];
2. Level measuring system readings [15,35];
3. Estimation of the density of the substance (product) stored in the tank [36,37].

The measurement process is influenced by the calibration of the tank, together with the level and density measurements.

As the measurement of product-level change (Figure 1) will affect the measurement uncertainty of the volume, modeling will focus on the following:

- Estimation of the absolute mass of the product in the tank;
- Estimation of product mass transfer (loading/unloading).

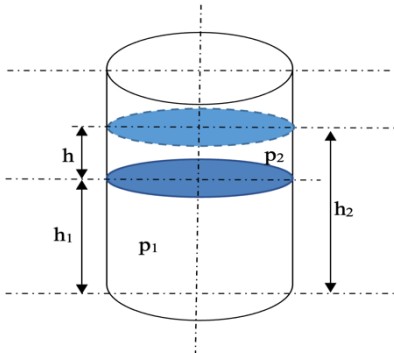

**Figure 1.** Measurement model of the change in liquid level in a vertical cylindrical tank.

### 2.1. Evaluation of the Product Mass in the Tank

For the calculation of the product mass, it was assumed that the temperature influence is negligible because this assumption will simplify the modeling, and the conclusions will not be affected. A well-known expression is used to relate the mass and volume:

$$m = V \cdot \rho \qquad (1)$$

where $V$ is the product volume, m$^3$; and $\rho$ is the density of the product, kg/m$^3$.

The density of the product in the container can be determined as follows:

1. Measured indirectly, using a level measuring system, at the actual temperature, $T_{act}$;
2. Estimated at the laboratory at a temperature, $T_0$, and then used for further calculations when the level measuring system does not have a density measurement function.

It should be emphasized that, as the temperature of the product changes, its volume and density also change. Thus, when the density is not measured and its value is obtained in the laboratory, it is necessary to evaluate the temperature changes of the product, that is, when $T_{act} \neq T_0$, to compensate for the thermal expansion [1,10,17].

During our analysis of scientific papers and standards, we noted that previous studies show that temperature has a dominant influence on tank volume [3,14,15,28–30]. A one-degree error causes a 0.03% error in the volume of fuel oil and a 0.05% error in the volume of crude oil [25]. This deviation can be achieved by using accurate temperature-measurement devices. Therefore, this error is classified as a systematic error and can be compensated [15] as stated in Formula (2):

$$m = V \cdot \rho \left[ 1 + \alpha (T - T_m) \right], \qquad (2)$$

where $T_m$ is the measured temperature of the oil product, $V$ is the measured volume of the oil product at temperature $T_m$, $\rho$ is the measured density of the oil product at temperature $T_m$, $\alpha$ is the volumetric temperature-expansion coefficient of the oil product, and $T$ is the temperature at which the mass of the oil product is calculated. Such temperature corrections are necessary according to standards [2,3,14]. Temperature error compensation

reduces the uncertainty related to temperature, and the component of total uncertainty can be expressed as $\frac{\partial m}{\partial \Delta T} \cdot u(\Delta T) = V \rho \alpha \cdot u(\Delta T)$, where $u(\Delta T) = \pm \frac{\Delta t}{\sqrt{3}}$, $\Delta t$ is the error of the thermometer. This component includes several variables that depend on the accuracy of the measuring instrument. Therefore, the temperature component can vary, and using commercially available thermometers, it can be reduced and made smaller compared to the other components of combined uncertainty for tank gauging [28,38,39]. Values of the coefficient for different materials can be obtained when calculated from specific volumes (densities) measured within a temperature range [40,41]. It is usually selected from the tables provided in the standards [42]. The thermal expansion coefficient decreases when the density increases and varies within a range of $\pm 5\%$ [43]. The uncertainty component of the coefficient can be expressed as $\frac{\partial m}{\partial \alpha} \cdot u(\alpha) = V \rho \Delta T \cdot u(\alpha)$, where $u(\alpha) = \pm \frac{\alpha \cdot 5\%}{100\% \cdot \sqrt{3}} = \pm \frac{\alpha}{20 \cdot \sqrt{3}}$. When $T = T_{\mathrm{m}}$, this component equals 0.

For further analysis, it was assumed that the density was measured by using a level measurement system. This method of analysis will better reflect the effect of all components on the combined uncertainty, without any assumptions about environmental/product temperature fluctuations, as both the product volume and density will be evaluated under the same conditions, that is, at $T_{act}$.

The most used density-measurement method for liquid products is based on pressure measurements in a liquid column. The measurement model can be expressed as a formula for the pressure, $P$, in a fluid:

$$P = \rho \cdot g \cdot h, \tag{3}$$

where $g$ is the gravitational acceleration, ~9.81 m/s$^2$; and $h$ is the product (liquid) height, m.

It is clear from Equation (3) that the assessment of the density of the product requires, in addition to the level measuring system, a pressure sensor, the error of which affects the assessment of the density and, as a result, the mass.

By combining (3) and (1), it is possible to express an indirect measurement model of the product mass inside the tank as follows:

$$m = V(h) \frac{P}{g \cdot h}. \tag{4}$$

The volume denoted in model (4), $V(h)$, is a function of the measured level $h$ and is determined from a graduation table built during tank calibration.

*2.2. Evaluation of the Uncertainty of Product Volume in the Tank Estimation*

The Guide to the Expression of Uncertainty in Measurement (GUM) [44] is applied to uncertainty assessment. Standard uncertainties that include the combined standard uncertainty are calculated as $u(y_i) = \sqrt{W_i^2 \cdot u(x_i)^2}$, where $W_i$ is the sensitivity coefficient, $W_i = \partial f / \partial x_i$, and $u(x_i)$ is the standard uncertainty that is evaluated by scientific judgment based on all of the available information on the possible variability of $x_i$. Each component is incorporated into a final expanded uncertainty: $U = k \times \sqrt{\sum_{i=1}^{N} u(y_i)^2}$. Expanded uncertainty is based on the standard uncertainty multiplied by a coverage factor of $k = 2$, which provides a confidence level of 95%.

Before discussing the simulation results, it is necessary to clarify what contributes to the uncertainty of volume estimation $u(V(h))$. The volume of the tank occupied by the liquid product was not modeled or measured; its value was taken from a table that was built during the calibration of the tank.

Calibration is a set of operations carried out to establish, under specified conditions, the relationship between the liquid level in the tank and the volume of that liquid [1]. One of the most important metrological characteristics is the maximum permissible uncertainty calculated according to the GUM [44]. It must qualify as $\pm 0.2\%$ of the indicated volume for vertical cylindrical tanks [1]. During the metrological supervision procedure, a calibration table, also known as a graduation table, is built: the minimum and maximum measurement

limits are determined, and the expanded uncertainties are calculated, which must comply with the maximum permissible value [3,16]. Thus, the measurement limits of the tank volume were not declared in advance but were selected based on the calibration results obtained at that time. Thus, they are not constant and may be subject to change after recalibration in service.

The mentioned uncertainty is influenced by two factors:

- Tank calibration procedure, i.e., the uncertainty of the tank graduation table;
- Level measurement: Even with an ideal tank graduation table, the level measurement is the variable that determines which row is considered as a volume estimation outcome.

Summarizing these statements, $u(V(h))$ can be expressed as follows:

$$u(V(h)) = \sqrt{u(V_{table}(h))^2 + u(V_h(h))^2},$$ (5)

where $u(V_{table}(h))$ is the standard uncertainty of the tank graduation table (in absolute form) at the estimated volume value, and $u(V_h(h))$ is the standard uncertainty of the tank volume estimate (in absolute form) due to the error of the level measuring system at the estimated volume value.

Modern tank-level measuring instruments, for example, radars, ensure a level measurement error of approximately ±0.5 mm. Such an error of level measurement results in relatively small change in the volume. The analysis of vertical tanks of a particular company showed that the influence of a level measurement error at the low filling levels of the tank is the greatest, and the uncertainty term is approximately 0.08% (varying slightly from tank to tank) and is dependent on the tank diameter. However, this uncertainty term decreases with an increase in the tank filling when the tank is full. Thus, when comparing this term (worst scenario—0.08%) to the uncertainty of the calibration table of the tank, which could reach up to 0.2% [1], it became obvious that the calibration uncertainty of the tank predominates in the volume estimation, and the influence of the level measurement system is small.

### 2.3. Evaluation of the Uncertainty of Product Mass in the Tank Estimation

Let us analyze the measurement uncertainty of the product mass in the tank and its components. The standard uncertainty, $u(m)$, when the gravitational acceleration value is considered an error-free constant (see measurement model Formula (4)), can be expressed as follows, as there are no indications about presence of correlation between input variables:

$$u(m) = \sqrt{[W(V(h){\cdot}u(V(h))]^2 + [W(P){\cdot}u(P)]^2 + [W(h){\cdot}u(h)]^2},$$ (6)

where $W(\ldots)$ is the sensitivity (or influence) coefficient of the input quantity, revealing the impact on the combined uncertainty; and $u(m)$ and $u(\ldots)$ are the standard uncertainties of the input quantities.

The influence coefficients are found as partial derivatives of the measurement model with respect to the corresponding input variable [44]. The derivation of these coefficients for volume, pressure, and level is given in expressions (7)–(9).

$$W(V(h)) = \frac{\partial m}{\partial V(h)} = \frac{P}{g{\cdot}h},$$ (7)

$$W(P) = \frac{\partial m}{\partial P} = \frac{V(h)}{g{\cdot}h},$$ (8)

$$W(h) = \frac{\partial m}{\partial h} = -V(h)\frac{P}{g{\cdot}h^2}.$$ (9)

By substituting (7)–(9) into (5), the expanded uncertainty (absolute form) of the mass measurement in the tank can be expressed as follows:

$$
U(m) = 2 \cdot \sqrt{ \underbrace{\left[ \frac{P}{g \cdot h} \cdot u(V(h)) \right]^2}_{\text{Volume term}} + \underbrace{\underbrace{\left[ \frac{V(h)}{g \cdot h} \cdot u(P) \right]^2}_{\text{Pressure term}} + \underbrace{\left[ \left( -V(h) \frac{P}{g \cdot h^2} \right) \cdot u(h) \right]^2}_{\text{Level term}}}_{\text{Density term}} }
\tag{10}
$$

The assumption of Gaussian distribution of the final quantity (i.e., the mass) allows us to take the coverage factor to be equal to 2 when the coverage probability is $p = 0.95$. Finally, the relative expanded uncertainty for the mass can be expressed as follows:

$$
U_s(m) = \frac{U(m)}{m}.
\tag{11}
$$

Equations (10) and (11) allows us to evaluate the measurement uncertainty of the product mass and to analyze the influence of the accuracy of the input quantities' measuring instruments and the filling level on the mass measurement.

The data for modeling the mass evaluation uncertainty (according to the real cylindrical tank parameters) are listed in Table 1.

**Table 1.** The parameters for modeling the mass evaluation uncertainty.

| Parameter | Value |
|---|---|
| tank filling level | (0.6–11.2) m |
| tank volume | (690–18,000) m$^3$ |
| density of stored product | 840 kg/m$^3$ |
| pressure sensor measurement error | 0.1% |
| level measurement error | 0.5 mm |

As can be seen from Figure 2, the product mass measurement is dominated by tank calibration uncertainty; other uncertainty terms, such as density determination and level measurement, are relatively small. For example, at a level of 6 m, the uncertainty of product mass is equal to 0.19% when $U(V(h)) = 0.15\%$. In addition, at extremely low product levels in the tank, the pressure and level measurement errors have a greater influence but become irrelevant when filling level $h > 3$ m is reached. For example, at a level of 1 m, the uncertainty of mass evaluation is equal to 0.25% when $U(V(h)) = 0.15\%$. An increase in the influence of pressure and level components on the uncertainty of mass estimation is observed. It is not available to reduce measurement uncertainty at extremely low tank fill levels by using the described mass measurement approach, as it aims at the density evaluation and, therefore, involves additional pressure and level measurements. If there is a need of more accurate measurements at these product levels, other density evaluation ways should be applied.

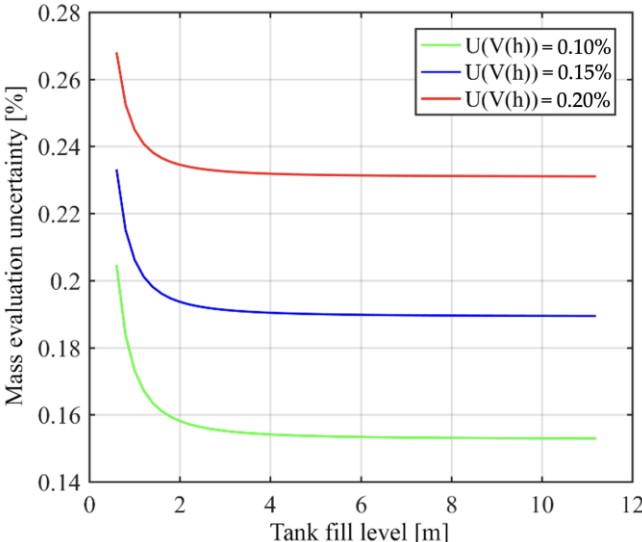

**Figure 2.** Dependence of the expanded uncertainty (relative form) of the product mass in the tank on the uncertainty of the tank graduation table ($U(V(h))$ is the expanded calibration uncertainty of the tank from the calibration certificate.

### 3. Estimation of the Uncertainty of Subtractive Measurements: Discussion

The total mass of the product in the tank is often not the desired result. In most cases, the change in product mass is more relevant, that is, the amount of product transferred to/from the tank. There are two methods to obtain this information:

(1) Use additional measuring instruments, e.g., flow meters;
(2) Perform two measurements of the product mass in the tank, namely before and after the transfer operation.

The first case falls outside of the scope of this study and is not be further analyzed in this paper. In the second case, when the change in mass is to be measured, it can be expressed as follows:

$$m_t = |m_2 - m_1|, \tag{12}$$

where $m_1$ and $m_2$ are the mass of the product before and after the transfer operation (transaction), respectively.

Because two mass measurements in the tank are performed to estimate the difference, each with its uncertainty, which is evaluated as discussed above, the combined uncertainty is expressed as the accumulation of these uncertainties. In other words, the uncertainty of the transaction mass, $m_t$, can be expressed by using the same approach as described in Section 2.2:

$$U(m_t) = 2 \cdot \sqrt{\left(\frac{U(m_1)}{2}\right)^2 + \left(\frac{U(m_2)}{2}\right)^2}, \tag{13}$$

where $U(m_1)$ and $U(m_2)$ are the expanded mass evaluation uncertainties that were obtained by using (10). The assumption of the Gaussian distribution of the mass quantity was made earlier; it remains true for assuming transfer mass (evaluated using (12)) distribution function. Therefore, the coverage factor is taken to be equal to 2 when the coverage probability is $p = 0.95$.

The relative form of the expression (13) would appear as follows:

$$U_s(m_t) = \frac{U(m_t)}{m_t} = \frac{\sqrt{U(m_1)^2 + U(m_2)^2}}{m_t}. \tag{14}$$

Another modeling experiment was performed for the same tank, using the parameters listed in Table 2.

**Table 2.** The input parameters for modeling of the mass transfer uncertainty.

| Parameter | Value |
|---|---|
| tank filling level | (0.6–11.2) m |
| tank volume | (690–18,000) m$^3$ |
| level change | (0.2–5) m |
| density of stored product | 840 kg/m$^3$ |
| pressure sensor measurement error | 0.1% |
| level measurement error | 0.5 mm |

The surface plot and its projection (Figure 3a,b) show that the uncertainty of a differential measurement depends highly on the initial tank filling and the change in the filling level during transfer. The extreme case is an almost full tank and a minor transfer (0.2 m change in filling level). In such a situation, the uncertainty can reach 20%. The difference in the masses ($m_2 - m_1$), which is equal to $m_t$, affects that for small transfers. The smaller the difference by which the expanded absolute uncertainty of the mass difference is divided, the larger the relative uncertainty. The uncertainty increases rapidly according to Formula (13), which describes the dependence of the mass difference uncertainty on the input parameters given in Table 2. However, as the amount of transferred product increases, the uncertainty of the estimation of the change in product weight decreases sharply, with a level change of 2 m, and the uncertainty decreases to 2%. It should also be noted that the relative uncertainty of the change in product mass depends linearly on the tank fill level present at the time of the transaction but nonlinearly on the transaction mass quantity.

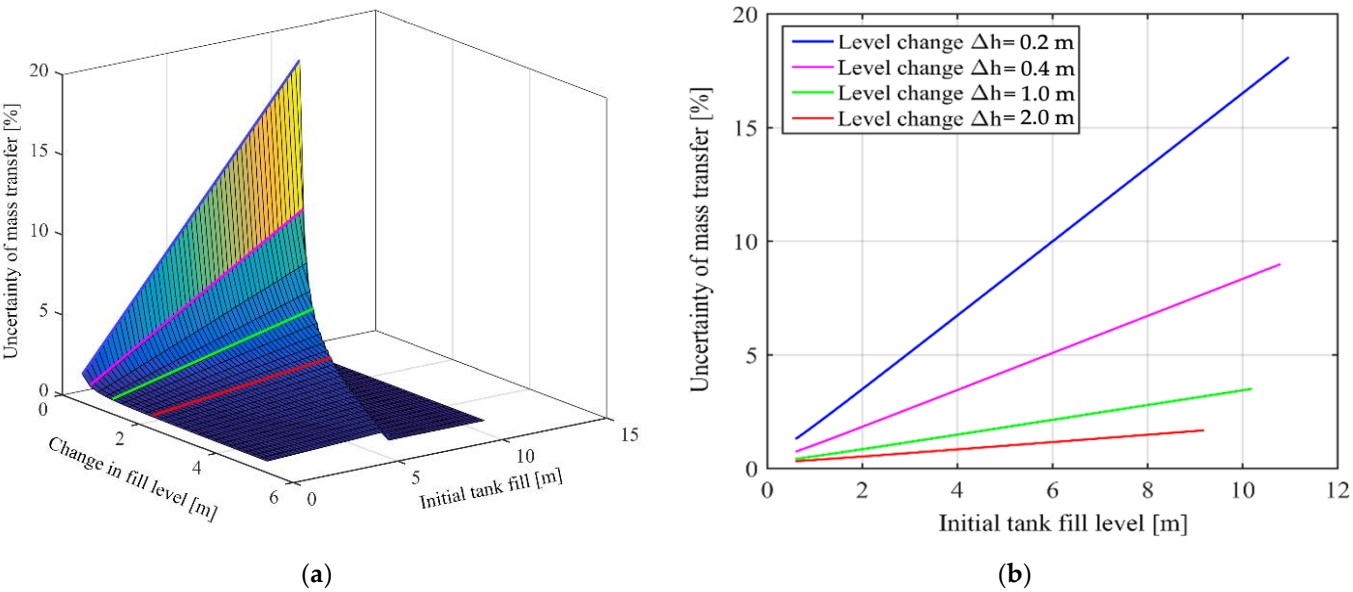

(**a**)   (**b**)

**Figure 3.** Dependence of the expanded uncertainty (relative form) of the transaction mass measurement on the tank filling level before transaction and the amount of the transaction mass: (**a**) surface plot; (**b**) surface plot projection at transaction mass values, namely 0.2, 0.4, 1, and 2 m.

In summary, the modeling results showed that tanks, as part of a product amount (volume) evaluation system, must be regularly maintained and inspected, as they make a large contribution to the final combined measurement uncertainty of product mass. The mass transfer to/from the tank is highly dependent on the initial amount of liquid product in the tank, as well as on the mass of the transaction. Knowing these trends, operators can adjust the workflow accordingly where/if possible. Therefore, the following is recommended:

- Custody and inventory transfer operations should be differentiated from those that measure the absolute inventory in a tank in real time. It is recommended to use mathematical expressions for the calculation of the change in mass between the two measuring points and the calculation of the uncertainty (Formulas (12), (13) and (14), respectively). The total standard uncertainty should be estimated from the standard uncertainties of the individual components of the mass difference. Note that the above formulas apply to the calibration results of the tanks at the normalized temperature. When measuring the volume of the tank, it is necessary to introduce temperature corrections for the specific case, owing to the thermal expansion. Otherwise, the temperature estimation uncertainty component can have a strong impact on the expanded system uncertainty [10,28,38].
- The filling/emptying processes of the existing tanks should be controlled, considering that the uncertainty of the measuring system is influenced by the workflow: (1) the fuller the tank, the higher the resulting measurement uncertainty; and (2) the higher the filling quantity, the lower the measurement uncertainty. The product transfer process must be organized in such a way that the one-time received and dispensed quantity at the same level of the tank would compensate each other. The product should be distributed on a tank farm based on the level at which the difference in mass is measured. If the amount of product transferred during an operation affects the level change in the tank by $\Delta h = 0.2$ m, the uncertainty of the mass of such a transaction changes linearly from 2% to 20% when the initial filling is 1 m $< h <$ 12 m. In the case of $\Delta h = 2$ m, the mass uncertainty in such a transaction changes linearly from 0.4% to 2.3% when the initial filling is 1 m $< h <$ 12 m.
- Mass and volume measurement uncertainties derived from absolute volume data are often overly optimistic. The uncertainty of the mass differential measurements is greater than it would be expected from the inventory measurements within the tank. This needs to be considered when determining the maximum measurement errors for all measuring instruments and/or measuring systems used in the accounting chain when designing and selecting measurement systems/equipment when forecasting worst-case scenarios.

## 4. Conclusions

This study examined the measurement uncertainty of the absolute and differential quantity measurement uncertainty by using tank gauging and their impact on loading/unloading processes.

The following was revealed by the study:

- The tank calibration uncertainty is the dominant term in mass evaluation in fuel tanks. Therefore, tanks, as part of the product amount evaluation system, must be maintained regularly and inspected. Other constituents in the uncertainty budget, namely density determination and level measurement, were relatively small. However, at extremely low product levels in the tank, the density and level measurement errors have a greater influence. This assumes that the weight in the tank is always corrected for temperature fluctuations if the product properties are set at a fixed (normalized) temperature.
- The uncertainty of the mass measurement transferred to/from the tank becomes highly dependent on the tank's initial fill level (i.e., before the transaction) and the amount of product transferred. If the tank is almost full and a small amount of product is transferred during operation, the uncertainty of the mass of such a transaction can be up to 20%. However, as the amount of transferred product increased, this uncertainty decreased sharply (up to 2% or less). For comparison, other authors have reported that typical manual tank gauging uncertainties range from 0.6% to 2.5% [28,45].
- Custody transfer operations should be separated from those intended to measure the absolute inventory in the tank, and a mathematical model related to the mass difference calculation was used. The mass of the contained product was evaluated

indirectly by measuring the fill level, tank geometric parameters, liquid temperature, and density.

- Although static tank measurements are less efficient than dynamic measurements realized with counters [28], the formulated recommendations allow for the management of custody transfer and can play a serious role in the amount of trust the operator can place in his inventory management system. This will help us understand that inventory management can indirectly control inventory accuracy. Larger measurement uncertainties in custody transfer result in higher losses and finances.

The provisions of this article are considerable for tank farm operators, such as refineries, chemical plants, and terminals, where an inventory of products and internal and external product transfer accounting is performed. By managing the filling/emptying processes of tank farms, direct financial losses can be reduced. These data can be used for the development of new inventory management software, as well as for process configuration. The proposed uncertainty evaluation can be used as a tool in decision-making under the application of liquid quantity measurement for the investigation of cylindrical tanks. It is needed for creating reliable testing, inspecting, and certification standards. The recommendations can be employed as a guide to estimate the reliability of the resulting models.

**Author Contributions:** Conceptualisation, A.M., P.K., and J.D.; supervision, A.M.; methodology, P.K., A.M.; software P.K.; investigation B.G.U., G.B.; formal analysis, J.D.; resources, G.B., J.D., and B.G.U.; writing—original draft preparation, B.G.U.; writing—review and editing, P.K., J.D., A.M. visualisation, P.K. All authors contributed equally. All authors have read and agreed to the published version of the manuscript.

**Funding:** This research received no external funding.

**Institutional Review Board Statement:** Not applicable.

**Conflicts of Interest:** The authors declare no conflict of interest.

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
