# Peer review of "Analysis of Liquid Quantity Measurement in Loading/Unloading Processes in Cylindrical Tanks"

_computation, doi:10.3390/computation10070122_

Round 1
Reviewer 1 Report
Comments and Recomendations:
1. All formulas in the article are numbered. But some of them are not mentioned in the text, for example, (1A),(3). It's not clear why formulas should be numbered if they are not referenced in the article.
2. The name of the institute is given four times in the title of the article. It would be easier and more aesthetically pleasing to write it once after the names of the authors, and indicate their e-mail addresses with numbers after.
3. Using formula (1A) to determine the temperature measurement error is not very productive at first glance. The coefficient of α is unlikely to have an accurate measured value and must also depend on temperature (to compensate for the absence of other terms of the used expansion). In the further text, its use was left by the authors without explanation. I think it makes sense to clarify the situation in the text.
4. The authors wrote about the practical significance of their article. But what is the scientific achievement of the work done? What scientific results of this work could be useful to other scientists in their research activities? This should be mentioned at least briefly, for example, in the conclusion.
This paper is well enough written to understand main results. The manuscript seems to be suitable for publication. I don't know exactly would such a work correspond to the content of the Journal of Computation or not. I incline to recommend it for publication there or in some other journal after minor mentioned corrections.
Reviewer 2 Report
The comments are in the manuscript.

Reviewer 3 Report
Nothing to report.
Reviewer 4 Report
As instruments in oil and its products amount measurement system chain, tanks must be regularly maintained and metrologically inspected because of significant measurement uncertainty. When measuring a change in the amount of stored material (i.e., transfer), the measurement uncertainty becomes highly dependent on the mass of the transaction and the initial liquid level in the tank. This work provides modeling of the uncertainties of the measuring system, which involves tanks, oil, and its product loading/unloading processes. It is shown that the volume/mass measurement accuracy depends not only on the tank calibration table but also on the accuracy of measuring instruments used and on the level of the liquid at the moment of measurement. The relative uncertainty of the change in product mass measurement depends linearly on the tank fill level present at the transaction time but nonlinearly on the mass transaction quantity. The manuscript belongs to the project report category and the following comments require attention.
1. Provide proper references for the equations.
2. State the reasons for a lower uncertainty for the fluid level height lesser than 2m.
3. Explain the uncertainty relations and values for better understanding.
4. Improve the English and grammar.
5. Include the further scope.
Author Response
Please see the attachment.

This manuscript is a resubmission of an earlier submission. The following is a list of the peer review reports and author responses from that submission.
Round 1
Reviewer 1 Report
Estimating measurement uncertainty is not a simple task, but it is necessary for many areas. Measurements are made in every area: medicine, industry, environmental protection, and many others. Therefore, students in the first years of study, especially in technical faculties, are familiarized with the measurement uncertainties. This article is at the engineering level and deals with the basic issues that students in the first years of study are acquainted with. Engineering designs are sometimes more advanced than what the authors presented. The conclusions drawn by the authors are “bleeding obvious”. It is obvious that the relative measurement uncertainty will increase with decreasing the volume/level/mass of the stored substance. It is also obvious that the double measurement of the mass (before and after the change of the level) will introduce twice the uncertainty of the measurement of this mass. In addition, tank calibration tables are being slowly abandoned.
The article mentions the thermal expansion of materials, but there is no evidence in the article about the influence of this parameter on the measurement uncertainty. The dynamics of filling or discharging the substance into/from the reservoir were also not considered. The basic problem is not only the density, but also the viscosity of the substance (depending on the temperature), and thus the amount of the substance "stuck" to the walls of the tank. The authors do not even mention it.
In addition, the authors' use of different symbols in the equations and others in the descriptions is blatant. It should be emphasized that the symbol of the measured quantity is not only a letter but also whether it is small or large, as well as straighten, tilted or even bold. Having GUM and VIM at my disposal, I suggest the authors get acquainted with this issue. I am ignoring the fact that it is necessary to correct numerous linguistic and editorial errors appearing in the text.
To sum up: despite the common necessity to estimate uncertainty and its significance in settlements between the client and the entrepreneur, this article does not contain any knowledge that would bring any novelty to the area of metrology. The article, after editorial corrections, should be printed in trade magazines intended for producers and users of tanks.
Reviewer 2 Report
The authors present an interesting project report on liquid quantity measurement in loading and unloading process in cylindrical tanks, also providing the uncertainty assessment approach.
Both on a scientific/technical and commercial approach, it is crucial to define all the uncertainty components to assess the measurement accuracy. Thus, the contents of the manuscript can draw a lot of attention from several research groups as it’s not always straightforward to develop the correct model to assess the measurement uncertainty. However, few minor revisions are needed before it can be accepted:
Minor revision
I suggest authors to revise the English language, especially for some sentences
Page 2, line 53
Please, replace m3 with m3
Figures 2, 3
I suggest authors to include the unit of measurement within square brackets
Page 6, paragraph 3
To clarify and explain the 20% uncertainty reported in line 220, making it immediately clear for any reader, it could be useful to express the relative uncertainty. In this way the difference in the masses (m2-m1) would be the denominator and therefore it would be immediately evident that for small transfers the uncertainty increases rapidly.
Page 7, lines 253, 255
A space is needed between the numbers and the unit of measurement
Page 8, lines 277
Please, replace “in-creases” with “increases”
Reviewer 3 Report
This paper provides modelling of uncertainties of measuring system involving tanks, in oil and its products loading and unloading processes. It is shown that the accuracy of volume / mass measurement depends not only on the tank calibration table, but also on the accuracy of other measuring instruments used and on the level of the liquid while measurement. The relative uncertainty of measurement of the change in product mass depends linearly on the tank fill level present at the time of the transaction, but nonlinearly on the transaction mass quantity
The manuscript requires the following comments to be addressed.
- Poor literature. There is no recent literature. No literature from 2022 and only two from 2021. Refer and cite the recent literature.
- Cite the proper sources for the equations.
- State the novelty and research gap at the end of the Introduction clearly and precisely.
- Improve the conclusion section with the significant numerical results using bulletin points.
- Provide the further scope.
- Provide a nomenclature
- Correct the units properly for the superscripts.
- Avoid clustered references, e.g. [9-13].
- Compare the results with literature/other measurement methods.
- Overall, English and grammar require improvement.
Round 2
Reviewer 1 Report
In the instructions for Authors we read:
"Types of Publications
Manuscripts submitted to Applied Sciences should neither be published previously nor be under consideration for publication in another journal. [...]
Articles: Original research manuscripts. The journal considers all original research manuscripts provided that the work reports scientifically sound experiments and provides a substantial amount of new information. "
I do not find anything new to the world of science in this article. The authors write about things that are known and obvious, although they are certainly useful. The article does not meet the basic requirements for this journal.
Reviewer 3 Report
The comments are addressed appropriately.